# Image Analysis of Adventitious Root Quality in Wild Sage and Glossy Abelia Cuttings after Application of Different Indole-3-Butyric Acid Concentrations

**DOI:** 10.3390/plants11030290

**Published:** 2022-01-21

**Authors:** Danilo Loconsole, Giuseppe Cristiano, Barbara De Lucia

**Affiliations:** Department of Agricultural and Environmental Sciences (DiSAAT), University of Bari “Aldo Moro”, 70126 Bari, Italy; danilo.loconsole@uniba.it (D.L.); barbara.delucia@uniba.it (B.D.L.)

**Keywords:** *Abelia* *x* *grandiflora*, exogenous plant growth regulators, *Lantana camara*, root morphological characterization, woody plants

## Abstract

Adventitious root (AR) formation is a key step in stem cutting propagation of economically important woody ornamentals. Inadequate environmental and hormonal conditions can lead to the production of an insufficient or modest number of ARs in stem cutting, with a consequent decrease in quality. The aim of this research was to optimize wild sage and glossy abelia autumn stem cutting propagation protocols, using image analysis to assess the effects of different IBA concentrations and cultivars on AR quality. For both taxa, the treatments were: four IBA concentrations: 0, 1250, 2500 and 5000 mg L^−1^ and two cultivars: ‘Little Lucky’ (cv1) and ‘Yellow’ (cv2) from Lantana, and ‘Canyon Creek’ (cv1) and ‘Eduard Goucher’ (cv2) from Abelia. Results show that IBA application is not needed to enhance rooting ability; however, IBA concentration is an important factor determining the best overall AR quality in both taxa. In wild sage applying 5000 mg L^−1^ IBA improved AR quality in ‘Little Lucky’, increasing the root number, total length, surface area and number of forks and crossings, but decreased quality in ‘Yellow’. In glossy abelia ‘Edouard Goucher’, 5000 mg L^−1^ IBA increased the root number, but 1250 mg L^−1^ IBA improved AR quality; ‘Canyon Creek’ did not perform as well as cv2 at these concentrations. This study confirms that sensitivity to IBA dosage varies among species and their cultivars. Findings may help the commercial nursery industry produce higher quality cuttings.

## 1. Introduction

Woody species can be propagated using two different methods: vegetative multiplication, which produces clones, and sexual reproduction [1,2]. Seed germination may be poor or erratic due to dormancy [3]; moreover, the viability of seeds is very low [4]. In the horticultural industry, vegetative propagation is the most popular method, as it is cheaper, easier and faster than sexual reproduction [5]; the plants produced are clones that maintain the morpho-physiological and genetic characteristics of the stock plants, uniformity and earliness of production [6].

A number of techniques may be adopted in the vegetative propagation of woody plants. Although layering is a simple technique, it is expensive and produces a small number of clones [7]. Grafting guarantees adaptability to unfavorable pedoclimatic conditions and resistance to telluric pathogens and parasites, but it is costly and labor-intensive, and compatibility problems may arise between graft and rootstock [8,9,10]. In-vitro propagation allows the production of a large number of plants in a very short time [11,12]; however, this technique requires equipped laboratories and skilled labor [13]. Moreover, many woody species are recalcitrant to adventitious rooting from stem cuttings, even if they are micro-propagated [14]. Propagation by stem cutting, less expensive and easier than in-vitro propagation, is the method most widely used to propagate clones of forest, fruit and shrub species [15,16,17].

Adventitious root (AR) formation is a key step in stem cutting propagation [18,19] of economically important woody ornamentals [20]. Inadequate environmental and hormonal conditions can lead to the production of an insufficient or modest number of AR in the stem cutting, with consequent loss of quality. Moreover, AR quality is characterized by a high number of very fine roots, which are essential for continuous access to water and nutrients [21] and can help the plant withstand transplant shock, increasing survival and plant growth [22,23,24] and by an adequate number of higher calibre roots with mechanical support function. Generally, the application of specific hormones to woody species can enhance AR quality, as number of roots and length development [25,26]; moreover, they can improve the plant material in so far that the cuttings develop more roots in a shorter time. Rooting hormones can improve rooting in some species but may have little effect on others [27]; furthermore, the specific stimulant and its concentration can affect uptake and subsequent adventitious rooting of stem cuttings: indole-3-butyric acid (IBA) is the most widely used auxin in the ornamental nursery industry [28].

In recent years, commercial ornamental nurseries in the European and Mediterranean Plant Protection Organization (EPPO) region have shown considerable interest in successful vegetative protocols for taxa that do not host *Xylella* bacterium, such as *Lantana* and *Abelia*.

*Lantana camara* L. (*Verbenaceae*), also known as wild sage, is native to tropical, subtropical and temperate regions [29]; it is a small vigorous shrub [30] of ornamental interest due to its attractive dense foliage and multi-colored flowers. Low-maintenance cultivars such as ‘Little Lucky Red’ (Ball Flora Plant) and ‘Bandana Yellow’ (Syngenta) are perfect for pot-tight production.

*Abelia x grandiflora* (Andrè) Rehd., (*Caprifoliaceae*) as glossy abelia, a hybrid between *A. chinensis* and *A. uniflora* [31], is a semi-evergreen, sprawling shrub with red-tinged leaves and multiple stems. It is used as a shrub border or hedge. A limited number of cultivars of *A. x grandiflora* are commercially available [32], among which the following are cultivated in Mediterranean countries: ‘Canyon Creek’ (*A. chinensis x A. x grandiflora*) and *A.* ‘Edward Goucher’ (*A*. *x grandiflora x schumanii*). Commercially, *A. x grandiflora* is propagated through semi-hardwood cuttings.

For these taxa, the literature dealing with IBA concentrations in relation to AR quality is still scarce and confusing, and further study is required to improve AR quality. Wild sage may asexually reproduce through stem softwood cuttings in spring or summer [33]. Researchers at the University of Florida found that semi-woody cuttings taken at the beginning of summer and treated with 8000 ppm IBA rooted successfully [34]. Instead, Blythe et al. [35] found that cuttings taken at the end of May and treated with 1000 ppm potassium salt of IBA yielded the best rooting results.

Moreover, despite its importance, AR quality and morphology are rarely considered in ornamental woody plants research applying image analysis software.

The aim of this study was to optimize wild sage and glossy abelia autumn stem cutting propagation protocols, using image analysis to assess the effects of different IBA concentrations and cultivars on AR quality.

## 2. Results

In general, the results of this study clearly show that vegetative propagation of wild sage and glossy abelia via semi-hardwood stem cuttings taken in early autumn is possible. Under our experimental conditions, ANOVA revealed no statistically significant interaction among IBA concentration and cultivar for rooting ability (%) in both taxa, with absolute values higher than 80%. Synthetic hormone application is not required to enhance rooting ability, although it does improve AR quality.

The interaction effect of IBA concentration and cultivar on the number of roots per cutting was significant at *p* ≤ 0.05. Table 1 shows that in both wild sage cultivars at 20 DAC, the best response was obtained by applying 5000 mg L^−1^ IBA (cv1 15 roots and cv2 17 roots); in glossy abelia, cv2 produced the highest number of roots per cutting (27) when treated with 5000 mg L^−1^ IBA, whereas the same dose produced a 27% decrease in cv1 (19.6 roots).

When the IBA concentration × cultivar interaction was examined for significance, all AR morphological traits that make up the quality, significantly differed at the 5% level, suggesting that the effects of IBA treatments on root length, surface area, diameter and on number of tips, forks and crossings differed with variety.

As for total root length in wild sage, the interaction effect of IBA concentration and cultivar was significant at the three sampling dates (Figure 1A,B). At 20 DAC, the highest value (366 mm) was recorded in cv1 treated with 5000 mg L^−1^ IBA; in contrast, the same dose had little effect on cv2 root length (158 mm). In the interaction effect, cv2 cuttings treated with 1250 mg L^−1^ IBA reached a length of only 208 mm (Figure 1B). At both 35 and 50 DAC, cv1 cuttings treated with 5000 mg L^−1^ IBA reached statistically significant maximum values, at 433 (+33%) and 748 mm (+105%), respectively, compared to cv2. When treated with the same dose, cv2 cuttings reached a root length of 325 mm (35 DAC) and 364 mm (50 DAC). The total root length value, at 50 DAC, of glossy abelia treated with 1250 mg L^−1^ IBA was significantly higher at twice that in cv2 (1624 mm) as compared to cv1 (861 mm) (Figure 1C,D).

In the interaction effect, at the three sampling dates, 5000 mg L^−1^ IBA concentration applied to wild sage cv1 (Figure 2A) showed the statistically highest root surface area values (76, 105 and 113 mm^2^, respectively); at the same concentration (Figure 2B), cv2 values were always lower than cv1 values (44, 63 and 74 mm^2^, respectively). Therefore, at the end of cutting cycle, at 5000 mg L^−1^ IBA, cv1 performance increased by 53% compared to cv2. IBA x cultivar interaction significantly affected surface area in glossy abelia: the largest root surface area was obtained in cv2 treated with 1250 mg L^−1^ IBA (Figure 2D): 135 and 134 mm^2^ at 20 and 35 DAC respectively; at the same concentration (Figure 2C), cv1 values were always lower than cv2 values (51 and 114 mm^2^ respectively). At 50 DAC, 1250 and 5000 mg L^−1^ IBA doses applied to cv2 cuttings had the highest root surface area values (205 and 202 mm^2^ respectively); in contrast, the same concentrations caused cv1 cuttings to decrease by 28 and 37% (Figure 2C).

As for root diameter (Table 2) in wild sage, interaction was not significant at three sampling dates; in glossy abelia, at 50 DAC, the 5000 mg L^−1^ IBA concentration had the same effectiveness in both cultivars, showing a maximum value of 0.57 mm in cv1 and of 0.54 mm in cv2. In this cultivar, the 1250 mg L^−1^ IBA dose had the smallest diameter.

In our experiment, IBA concentration and cultivars interaction significantly affected root tips number per cutting in wild sage and glossy abelia at 20, 35 and 50 DAC (Table 3). At 20 DAC, 5000 mg L^−1^ IBA was the most effective dose in both wild sage cultivars, although cv1 showed a better response (111 tips), differing statistically from cv2 (73 tips). At 35 DAC, 5000 mg L^−1^ IBA was again the optimal dose in both cultivars (cv1 103 tips and cv2 95 tips). At the end of the rooting cycle (50 DAC), the effectiveness of the 5000 mg L^−1^ IBA dose was confirmed in both cultivars, respectively with 272 tips in cv1 and 335 tips in cv2. As for glossy abelia, the cv2 cuttings, treated with 1250 mg L^−1^ IBA, reached statistically different and higher number of tips compared to the other treatments: 401 at 20 DAC, 542 at 35 DAC, 819 tips at 50 DAC. On the contrary cv1, at 50 DAC, cv1 showed a 58% decrease compared to cv2 with the same dose.

As expected, significant differences were observed in IBA x cultivar interaction with respect to the number of root forks (Table 4). At 20 DAC, the application of IBA to both wild sage cultivars resulted in a greater number of forks than in the untreated control. Values for cv1 cuttings treated with IBA 5000 mg L^−1^ (385 forks) were more than double those for cv2 (152 forks). At 35 and 50 DAC, cv1 treated with IBA 5000 mg L^−1^ showed comparable strong effects at 458 (+103%) and 570 forks (+125%), respectively, compared to cv2. At 20 DAC, 1250 mg L^−1^ IBA applied to cv2 glossy abelia cuttings influenced the number of forks significantly (566); in contrast, the same IBA dose applied to cv1 results in only 207 forks. This trend was maintained at 35 DAC, and it was confirmed at 50 DAC: 1250 mg L^−1^ IBA produced only 398 forks in cv1 (−63%), but a significantly higher value of 1078 forks in cv2 (Table 4).

The interaction effect of IBA concentration and cultivar with respect to the number of root crossings is shown at the three sampling dates in Table 5. Results highlight differences among the cultivars when 5000 mg L^−1^ IBA was applied to the wild sage cuttings: cv1 showed the highest number of crossings, at 45, 49 and 90, respectively, with increases compared to the cv2 at 66%, +22% and +43%, respectively; cv2 shows statistically different, poorer performances (27, 40 and 63 crossings), although still better with respect to the other treatments. As for glossy abelia, the cv2 cuttings treated with the 1250 mg L^−1^ IBA dose already showed high values (198 crossings at 50 DAC). In contrast, when the same dose was applied to cv1, only 71 crossings (−64%) were obtained.

When the proportions of total root length were placed into different diameter classes, in wild sage, compared to all other concentrations, cuttings treated with 5000 mg L^−1^ IBA had a significantly higher proportion in the very fine root diameter class (0–0.5 mm) at 35 DAC (Figure 3C). Cuttings treated with 2500 and 5000 mg L^−1^ IBA showed a higher proportion of total root length in the very fine root diameter class (0–0.5 mm) at both 20 and 50 DAC (Figure 3A,E).

In glossy abelia, most total root length proportions fell in the very fine (0–0.5 mm) diameter class, and distinct patterns were observed among IBA concentrations at 20 (Figure 3B), 35 (Figure 3D) and 50 DAC (Figure 3F). In the very fine (0–0.5 mm) diameter class, total root length values were significantly higher at 1250 mg L^−1^ IBA than at other concentrations. For large root diameter classes (>1.0 mm), different IBA concentrations yielded similar proportions of total root length at the three sampling dates (Figure 3B,D,F).

## 3. Discussion

This study aimed to optimize wild sage and glossy abelia autumn stem cutting propagation protocols, using image analysis to assess the effects of different IBA concentrations and cultivars on AR quality. Woody species are clonally multiplied through different methods, including root cuttings [4]. Cutting success, entailing quality AR formation [36] with high rooting percentage, depends on numerous factors such as cutting type, environmental conditions, nutritional levels of the stock plant, rooting medium and phytohormone application [37,38]. Several studies have highlighted the importance of auxin use in improving AR formation and maximizing production efficiency in woody species with high market demand [5,39,40,41]. Ornamental nurseries frequently apply IBA to cuttings, although not in concentrations tailored to the specific shrub species or cultivar; this leads to increased costs and often poor-quality root cuttings. Depending on the species and the cultivars, in accordance with [42,43,44], auxin treatment is known to have positive effects on rhizogenesis at specific concentrations. Understanding which is the most effective IBA concentration in terms of forming quality ARs is crucial to the cutting propagation industry [45].

Under our experimental conditions, wild sage and glossy abelia stem cuttings harvested in early autumn had high rooting percentages at 20 DAC, irrespective of exogenous IBA treatment. These results agree with those of [46], who determined that polar transport of endogenous auxin from the apical meristem to the nodes of sunflower cuttings is sufficient to stimulate AR formation without the addition of exogenous auxin.

Few studies have compared the morphological quality of AR in wild sage and glossy abelia cuttings treated with different IBA concentrations. In general, our study suggests that the effects of IBA concentrations differ from one cultivar to the other in each taxon. The interaction between IBA concentration and cultivar was significant for root number (Table 1). Our data show that the application of root stimulant can have a positive effect on the number of roots per cutting; however, while the 5000 mg L^−1^ concentration improved the number in both wild sage cultivars, the effect on glossy abelia cultivar 2 was 38% better (27 roots per cutting) than in glossy abelia cv1. Similar findings were reported by Dawa et al. [47] who found that when treated with 1500 ppm IBA, the ‘First Red’ rose cultivar yielded the maximum root number (19.0), whereas the ‘Naranga’ cultivar yielded the minimum root number (3.1). The differences between cultivars could be due to the genotype or the content of endogenous auxin in the cutting. IBA is one of the strong plant growth regulators in culture medium too for stimulating and increasing the number of roots. Samani et al. 2016, in Lantana showed that the highest number of roots was induced in MS media with 0.25 mg L^−1^ IBA [48].

At 50 DAC, our results on AR morphology in wild sage (Figure 1A,B and Figure 2A,B and Table 4 and Table 5) show that in cv1, compared to cv2, the highest IBA concentration (5000 mg L^−1^) improves the quality, which is more than satisfactory, for the character length (+105%), surface area (+53%), number of forks (+125%) and crossings (+43%). In contrast, at 50 DAC, applying a quarter of the maximum dose (1250 mg L^−1^) of IBA to glossy abelia cv2 yields root lengths that was double that of cv1 (Figure 1C,D). At 1250 mg L^−1^ IBA, cv2 also performs better than cv1 with respect to root surface area (+28%, Figure 2C,D), number of tips (+139%, Table 3), forks (+170%, Table 4) and crossings (+178%, Table 5). Previous research [49,50] has shown that, in some species and cultivars, auxins stimulate root initiation but disturb root elongation at increasing IBA concentrations. In our experiment, cv1 cuttings of glossy abelia showed very poor AR quality when treated at concentrations higher than 1250 mg L^−1^. Our results agree with [51], that *Veronica pusanensis* showed a similar tendency at high IBA concentrations (0.5 and 1.0 mg L^−1^). Under our experimental conditions, it could be concluded that IBA influenced root diameter too, although in a different way in both taxa (Table 2); similar data were recorded by [4] in Cotinus coggygria ‘Kanari’ and Syringa vulgaris ‘President Grevy’. Our results therefore confirm that sensitivity to IBA dosage varies among species and their cultivars. As argued by [52], the root fineness (Figure 3), recorded image analysis techniques, could be a very important trait for the water and nutrient uptake in woody cuttings.

## 4. Materials and Methods

### 4.1. Plant Materials and Rooting Environment

Semi-hardwood stem cuttings from wild sage and glossy abelia landscape stock plants were used as material for this study. Experiments were designed separately in the two studied taxa and they were conducted from 1 October to 20 November 2020 at an ornamental nursery farm situated in Monopoli (Bari, Italy, 40°54′19.1″ N, 17°18′21.4″ E; 66 m above sea level) under an ethylene–vinyl acetate propagation greenhouse with 50% shading.

For this study, 18 median cuttings were taken randomly from a single clonal mother plant, in each taxon and cultivar, making sure that they were disease-free, moderately vigorous, true-to-type, and uniform. Each cutting was 5–6 cm long and had three nodes. Leaves on the bottom half of the cuttings were removed, and the remaining leaves were cut back by 50%. The basal end of the stem cutting was cut at an angle to maximize the absorbing surface for effective rooting. It was dipped into the selected IBA concentration treatments (epidermal application to 10 mm of basal stem) and immediately planted into prepared seedling trays.

To plant cuttings, sanitized plastic trays (104 holes and 3.5 cm in diameter) were filled with a high-quality commercial growing medium for cuttings (pH, 5.0–6.0; organic carbon, 35%; organic nitrogen, 0.8%; organic matter, 85%) consisting of a blend of perlite and brown and blond *Sphagnum* peat. The medium was well pressed in the cells and watered until saturation. Cuttings were treated and immediately planted into holes punched into the growing medium. Planted trays were transferred into a rooting greenhouse, treated with a systemic fungicide (Propamocarb) and then covered with nonwoven fabric for four days to reduce abiotic stress.

Rooting greenhouse environmental parameters were monitored: air temperature ranged from 12 °C (minimum night temperature) to 20 °C (maximum day temperature); the seedbeds (bottom heating) were heated to a temperature of 18 ± 1 °C, and a 60 s misting was scheduled every 20 min from 8 a.m. to 3 p.m. for the duration of the experiment.

### 4.2. Experimental Design

For both taxa, the treatments were:(i)Four IBA concentrations: 0 (control), 1250, 2500 and 5000 mg L^−1^ (Sigma, St. Louis, MO, USA); control (0 mg L^−1^ IBA) cuttings were soaked in distilled water;(ii)Two cultivars: ‘Little Lucky’ (marked as cv1) and ‘Yellow’ (cv2) from Lantana and ‘Canyon Creek’ (cv1) and ‘Eduard Goucher’ (cv2) from Abelia.

Treatments were delivered in randomized complete block design and 24 experimental units (4 concentrations × 2 cultivars × 3 replicates). Each experimental unit consisted of eighteen cuttings (*n* = 432 cuttings from each taxon).

### 4.3. Rooting Ability Measurements

The root number (roots ≥ 0.5 cm length) per cutting was determined at 20 DAC. Rooting and mortality rates (%) were assessed at the end of the rooting period (50 DAC). The rooting rate (%) was calculated as the number of cuttings with AR per total number of cuttings × 100. Percentage data were subjected to arcsine square root transformation before ANOVA analysis.

### 4.4. Root Image Analysis

To analyze the quality of the root system, samples were taken at three different times: 20, 35 and 50 DAC. Six cuttings per experimental unit were taken on each sampling date. The substrate was gently washed away from the roots using first a hot bath and then a brush. The roots were scanned at 400 dpi using an Epson v700 Perfection (Japan) scanner. The captured images were then processed using image analysis software (WinRHIZO v. 2005b©, Regent Instruments Inc., Québec, QC, Canada, www.regentinstruments.com (accessed on 27 November 2021)) to determine total root length, root surface area, root average diameter, root tips, root forks and root crossings. The distribution of root length among the following root diameter classes was also determined [53]: very fine (VF, 0–0.5 mm), fine (F, 0.5–1 mm) and large (L > 1 mm).

### 4.5. Statistical Analysis

Two-way ANOVA was performed within each species for each sample date (20, 35, and 50 DAC) to test the effects of IBA concentration (IBA), cultivar (CV), and IBA × CV interaction on rooting ability, the number of roots and their morphological features.

All the above data analyses were performed using SAS version 9.3 statistical software (SAS, 1999); treatment means were separated by the S.N.K. (Student Newman–Keuls) test (*p* ≤ 0.05).

## 5. Conclusions

This study investigated the effects of different IBA concentrations on the quality of cuttings from two cultivars of wild sage and glossy abelia. Results, which can be used to optimize propagation protocols, indicate that:(i)Stem cutting propagation in early autumn is possible;(ii)IBA application is not needed to enhance rooting ability;(iii)IBA concentration is an important factor determining the best overall AR quality in both taxa;(iv)In wild sage cv1 (‘Little Lucky’), 5000 mg L^−1^ IBA improved AR quality by increasing root number, length, surface area and number of forks and crossings; cv2 (‘Yellow’) did not perform as well as cv1;(v)In glossy abelia cv2 (‘Edouard Goucher’), 5000 mg L^−1^ IBA increased the root number, but only 1250 mg L^−1^ IBA was needed to improve AR quality; cv1 (‘Canyon Creek’) did not perform as well as cv2.

The results of this study, which confirm that sensitivity to IBA dosage varies among species and their cultivars, could be relevant to the production of high-quality cuttings in the commercial nursery industry.

## Figures and Tables

**Figure 1 plants-11-00290-f001:**
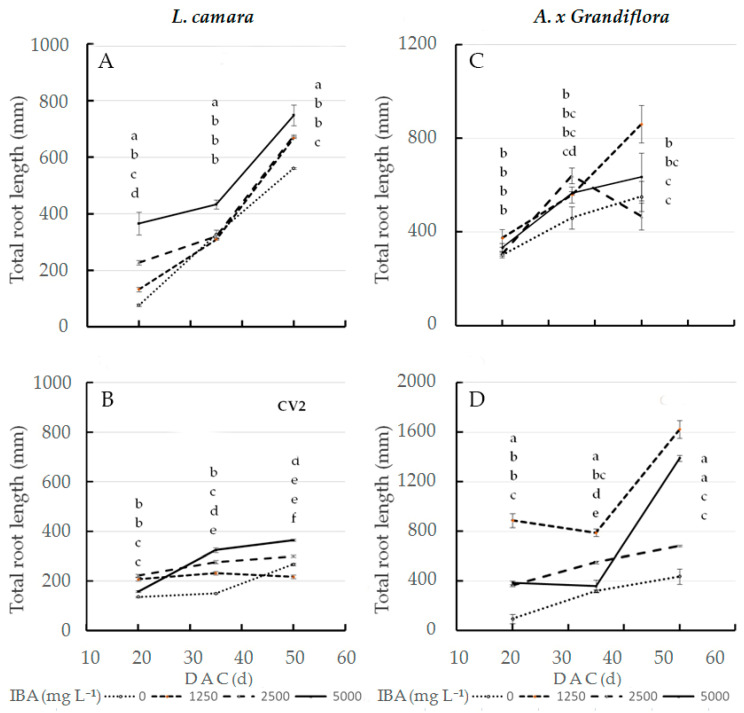
Total root length at three sampling dates in two *L. camara* cultivars ((**A**) ‘Little Lucky’, (**B**) ‘Yellow’) and in two *A. x grandiflora* cultivars ((**C**) ‘Canyon Creek’, (**D**) ‘Edouard Goucher’) treated with different IBA concentrations (*n* = 8). Different letters denote significant differences (*p* ≤ 0.05) among treatments for the same sampling date. Data shown are means ± SE of three replicates.

**Figure 2 plants-11-00290-f002:**
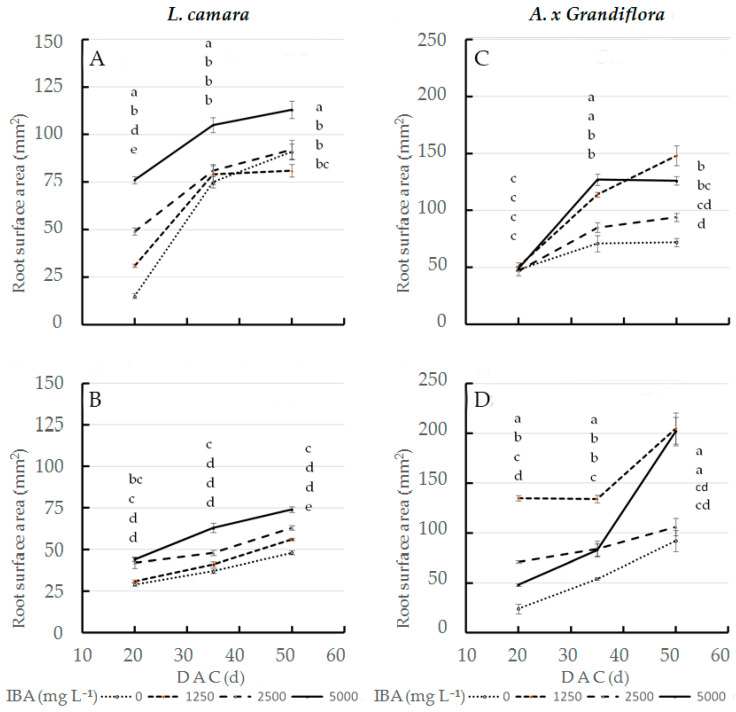
Root surface area at three sampling dates in two L. camara cultivars ((**A**) ‘Little Lucky’, (**B**) ‘Yellow’) and two *A.*
*x grandiflora* cultivars ((**C**) ‘Canyon Creek’, (**D**) ‘Edouard Goucher’) treated with different IBA concentrations (*n* = 8). Different letters denote significant differences (*p* ≤ 0.05) among treatments for the same sampling date. Data shown are means ± SE of three replicates.

**Figure 3 plants-11-00290-f003:**
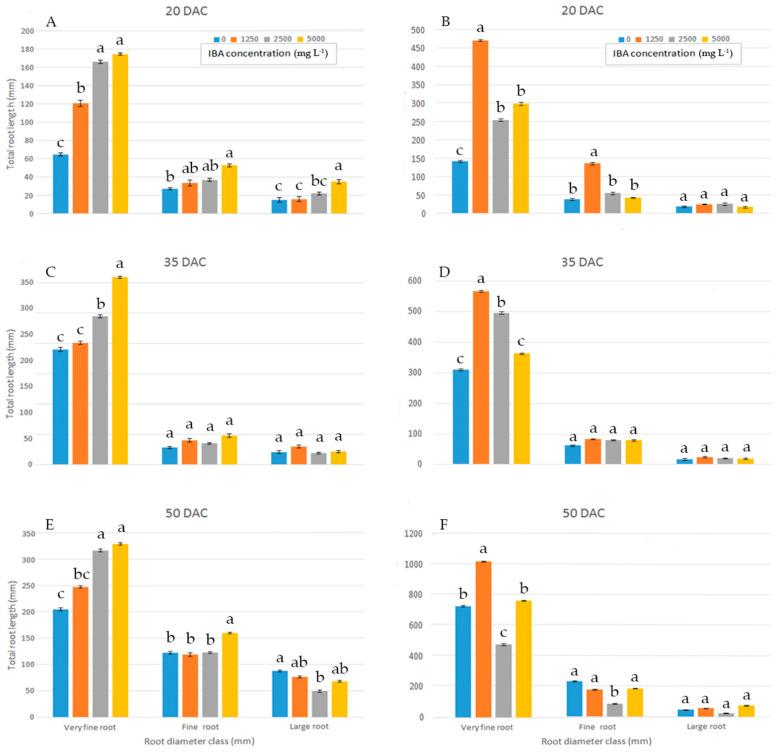
Total root length distribution (%) of wild sage (**A**,**C**,**E**) and glossy abelia (**B**,**D**,**F**) in three diameter classes, evaluated at three sampling dates, as effect of four IBA concentration. Vertical bars with different letters means significant differences according with S.N.K. test, *p* ≤ 0.05 for each class root diameter. Data shown are means ± SE of three replicates.

**Table 1 plants-11-00290-t001:** Number of roots per cutting in wild sage and glossy abelia at 20 DAC, according to IBA concentration and cultivar.

TMTS	Roots Per Cutting (No.)			
	*L. camara*		*A. x grandiflora*	
IBA (mg L^−1^)	cv1	cv2	cv1	cv2
0	3.0 ± 0.3 d	7.6 ± 1.2 c	6.6 ± 0.9 d	1.3 ± 0.3 e
1250	9.0 ± 0.6 bc	7.3 ± 0.7 c	9.3 ± 1.3 d	11.7 ± 1.2 c
2500	6.0 ± 0.6 c	11.0 ± 0.6 b	6.3 ± 0.9 d	23.0 ± 1.7 b
5000	15.0 ± 0.6 a	17.0 ± 1.1 a	19.6 ± 0.7 b	27.0 ± 1.5 a

*L. camara* cultivars: cv1 ‘Little Lucky’, cv2 ‘Yellow’; *A. x grandiflora* cultivars: cv1 ‘Canyon Creek’, cv2 ‘Edouard Goucher’. Within each taxon, data followed by the same letter are not significantly different at *p* ≤ 0.05 according to the Student Newman–Keuls (SNK) test. Data shown are means ± SE of three replicates. In this table and in the following tables and figures, DAC = Days After Cutting.

**Table 2 plants-11-00290-t002:** Root diameter in wild sage and glossy abelia at 50 DAC, according to IBA concentration and cultivar.

TMTS	Root Diameter (mm)			
	*L. camara*		*A. x grandiflora*	
IBA (mg L^−1^)	cv1	cv2	cv1	cv2
0	0.41 ± 0.03 a	0.33 ± 0.02 a	0.40 ± 0.03 cd	0.44 ± 0.02 cd
1250	0.49 ± 0.05 a	0.43 ± 0.02 a	0.49 ± 0.02 bc	0.37 ± 0.03 d
2500	0.46 ± 0.02 a	0.33 ± 0.01 a	0.51 ± 0.01 b	0.41 ± 0.03 cd
5000	0.42 ± 0.01 a	0.35 ± 0.02 a	0.57 ± 0.03 a	0.54 ± 0.02 a

*L. camara* cultivars: cv1 ‘Little Lucky’, cv2 ‘Yellow’; *A. x grandiflora* cultivars: cv1 ‘Canyon Creek’, cv2 ‘Edouard Goucher’. Within each taxon, data followed by the same letter are not significantly different at *p* ≤ 0.05 according to the Student Newman–Keuls (SNK) test. Data shown are means ± SE of three replicates.

**Table 3 plants-11-00290-t003:** Number of root tips per cutting in wild sage and glossy abelia at 20, 35 and 50 DAC, according to IBA concentration and cultivar.

TMTS	Root Tips (No.)											
	DAC											
	20		35		50		20		35		50	
	*L. camara*						*A. x grandiflora*					
IBA (mg L^−1^)	cv1	cv2	cv1	cv2	cv1	cv2	cv1	cv2	cv1	cv2	cv1	cv2
0	36 ± 1 e	35 ± 3 e	59 ± 7 bc	60 ± 1 bc	225 ± 20 b	141 ± 1 d	133 ± 6 e	92 ± 6 f	195 ± 27 c	310 ± 3 b	288 ± 15 d	649 ± 16 b
1250	38 ± 4 e	40 ± 1 e	51 ± 2 c	65 ± 2 bc	237 ± 17 b	175 ± 2 c	255 ± 10 bc	401 ± 41 a	295 ± 13 b	542 ± 13 a	343 ± 46 cd	819 ± 2 a
2500	57 ± 1 c	49 ± 2 d	61 ± 2 bc	75 ± 2 b	242 ± 20 b	198 ± 4 bc	180 ±8 de	264 ± 8 bc	322 ± 9 b	390 ± 7 b	452 ± 40 c	706 ± 12 b
5000	111 ± 3 a	73 ± 4 b	103 ± 5 a	95 ± 2 a	272 ± 15 ab	335 ± 4 a	324 ± 10 b	217 ± 10 cd	329 ± 8 b	366 ± 40 b	354 ± 40 cd	724 ± 5 b

*L. camara* cultivars: cv1 ‘Little Lucky’, cv2 ‘Yellow; *A. x grandiflora* cultivars: cv1 ‘Canyon Creek’, cv2 ‘Edouard Goucher’. Within each sampling date (20, 35 and 50 DAC) and for each taxon, data followed by the same letter are not significantly different at *p* ≤ 0.05 according to the Student Newman–Keuls (SNK) test. Data shown are means ± SE of three replicates.

**Table 4 plants-11-00290-t004:** Number of root forks per cutting in wild sage and glossy abelia at 20, 35 and 50 DAC, according to IBA concentration and cultivar.

TMTS	Root Forks (No.)											
	DAC											
	20		35		50		20		35		50	
	*L. camara*						*A. x grandiflora*					
IBA (mg L^−1^)	cv1	cv2	cv1	cv2	cv1	cv2	cv1	cv2	cv1	cv2	cv1	cv2
0	69 ± 4 f	93 ± 2 e	239 ± 8 c	130 ± 3 e	387 ± 10 c	175 ± 2 f	113 ± 9 d	89 ± 18 d	212 ± 15 c	102 ± 6 d	183 ± 44 e	683 ± 20 b
1250	133 ± 2 d	120 ± 3 d	353 ± 16 b	155 ± 1 de	384 ± 13 c	206 ± 2 ef	207 ± 16 c	566 ± 31 a	268 ± 5 b	367 ± 17 a	398 ± 11 d	1078 ± 23 a
2500	176 ± 6 b	129 ± 4 d	327 ± 20 b	174 ± 1 d	492 ± 16 b	227 ± 2 de	191 ± 28 c	357 ± 21 b	272 ± 33 b	372 ± 3 a	231 ± 44 e	576 ± 12 c
5000	385 ± 12 a	152 ± 3 c	458 ± 18 a	225 ± 1 c	570 ± 25 a	253 ± 2 d	201 ± 11 c	250 ± 15 c	407 ± 35 a	442 ± 15 a	375 ± 59 d	1142 ± 4 a

*L. camara* cultivars: cv1 ‘Little Lucky’, cv2 ‘Yellow; *A. x grandiflora* cultivars: cv1 ‘Canyon Creek’, cv2 ‘Edouard Goucher’. Within each sampling date (20, 35 and 50 DAC) and for each taxon, data followed by the same letter are not significantly different at *p* ≤ 0.05 according to the Student-Newman–Keuls (SNK) test. Data shown are means ± SE of three replicates.

**Table 5 plants-11-00290-t005:** Number of root crossings per cutting in wild sage and glossy abelia at 20, 35 and 50 DAC, according to IBA concentration and cultivar.

TMTS	Root Crossings (No.)											
	DAC											
	20		35		50		20		35		50	
	*L. camara*						*A. x grandiflora*					
IBA (mg L^−1^)	cv1	cv2	cv1	cv2	cv1	cv2	cv1	cv2	cv1	cv2	cv1	cv2
0	7 ± 1 e	10 ± 1 e	41 ± 3 b	21 ± 1 d	56 ± 4 b	31 ± 1 c	21 ± 2 c	12 ± 3 c	15 ± 1 e	23 ± 2 d	40 ± 7 d	117 ± 7 b
1250	11 ± 1 e	16 ± 1 d	23 ± 1 d	25 ± 1 d	54 ± 4 b	46 ± 2 b	27 ± 5 c	70 ± 5 a	37 ± 2 c	64 ± 3 a	71 ± 6 c	198 ± 4 a
2500	19 ± 1 cd	22 ± 1 c	35 ± 2 bc	29 ± 1 cd	88 ± 7 a	56 ± 1 b	20 ± 5 c	51 ± 5 b	58 ± 4 ab	49 ± 3 bc	62 ± 4 c	73 ± 5 c
5000	45 ± 3 a	27 ± 2 b	49 ± 3 a	40 ± 2 b	90 ± 8 a	63 ± 2 b	43 ± 2 b	19 ± 3 c	45 ± 5 bc	63 ± 6 a	66 ± 7 c	210 ± 7 a

*L. camara* cultivars: cv1 ‘Little Lucky’, cv2 ‘Yellow; *A. x grandiflora* cultivars: cv1 ‘Canyon Creek’, cv2 ‘Edouard Goucher’. Within each sampling date (20, 35 and 50 DAC) and for each taxon, data followed by the same letter are not significantly different at *p* ≤ 0.05 according to the Student Newman–Keuls (SNK) test. Data shown are means ± SE of three replicates.

## Data Availability

Data is contained within the article.

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
