# Peer review of "Image Analysis of Adventitious Root Quality in Wild Sage and Glossy Abelia Cuttings after Application of Different Indole-3-Butyric Acid Concentrations"

_plants, 2022, doi:10.3390/plants11030290_

Round 1

Reviewer 1 Report

The paper used the WinRhizo software to analyze the effects of IBA concentrations on rooting of sage and abelia cuttings. The experiment itself is good and the data is sufficient. However, WinRhizo was developed in 1995. The software has been widely used for root image analysis since then, including cuttings. So, there is no scientific significance to emphasis on using this software for root quality analysis of cuttings. And both wild sage and glossy abelia are easy to propagate by cuttings as a regular practice. So, in general, this paper did not resolve any scientific problems or technological difficulties. It is meaningless to conclude "This study has demonstrated that WinRHIZO technology can be used to analyze AR 399 morphology and quality in wild sage and glossy abelia cuttings treated with increasing 400 IBA concentrations." (Line 399-401) through such kind research. The results also did not significantly "help optimize early-autumn cutting production" as concluded (Line 406). 

For the writing, the introduction part is very disorganized. All the figures were repeated the same data in the tables (present as table or figure). All the data should be presented standard deviation or standard error. 

In general, regarding there is no significant scientific or technological problems of this paper can resolve or proposed, it is not suitable to publish on this journal. 

Author Response

The authors rewrote large portions of the manuscript

* Introduction was improved and it provides sufficient background. It now includes more relevant reference
* Results and Discussion were improved. They are clearly presented.
* Conclusions were improved. They are supported by the results

Reviewer 2 Report

It's an interesting and economically important topic, but nevertheless there are a number of open question:

l. 18 What does quality mean? Does it mean improved shoot development, a lot of fine roots or what do the authors mean with improved quality of the cuttings? Is there a margin, which seperates good quality from bad quality?

l 30-31 does this mean to improve the plant material in so far that the cuttings develope more roots in a shorter time?

l 48-49 I don't understand this sentence in this context, because the focus is vegetative reproduction; I agree that in vitro propagation needs specific equipement, but propagation of seeds seems to me another topic.

l55-60 these are very general statements, which can be shortened or deleted

l75 the term optimal is still unclear; It would have been good to combine the data with data of shoot development and/or mortality after transplanting, because this could give information about the vitality (quality?) of the  plant material

l85-94 this can be deleted, because it's not the focus of investigation

Fig 1 and 2 a scale is missing

Fig 5 and 6 can be combined to one figure (perhaps smaller subfigures)

l314-316 to me it's not confirmed, because stress and diseases were not investigated

Author Response

The authors rewrote large portions of the manuscript

  • Introduction was improved and it provides sufficient background. It now includes more relevant reference
  • Results under discussion were improved. They are clearly presented.
  • Conclusions were improved. They are supported by the results

  1. 18 What does quality mean? Does it mean improved shoot development, a lot of fine roots or what do the authors mean with improved quality of the cuttings? Is there a margin, which seperates good quality from bad quality?

Lines 54-60

l 30-31 does this mean to improve the plant material in so far that the cuttings develope more roots in a shorter time?

Lines 62-63

l 48-49 I don't understand this sentence in this context, because the focus is vegetative reproduction; I agree that in vitro propagation needs specific equipement, but propagation of seeds seems to me another topic.

This sentence was deleted

l55-60 these are very general statements, which can be shortened or deleted

This sentence was deleted

l75 the term optimal is still unclear; It would have been good to combine the data with data of shoot development and/or mortality after transplanting, because this could give information about the vitality (quality?) of the  plant material

This  term was deleted

l85-94 this can be deleted, because it's not the focus of investigation

This sentence was deleted

Fig 1 and 2 a scale is missing

Figures were deleted

Fig 5 and 6 can be combined to one figure (perhaps smaller subfigures)

Figures ex 5-6 were combined to one Figure:, now Figure 3

l314-316 to me it's not confirmed, because stress and diseases were not investigated

This sentence was deleted

Reviewer 3 Report

This manuscript describes results from a practically-oriented study aimed at assessing optimum concentration of growth regulators for vegetative propagation of woody plants by stem cuttings.

Title

Do not capitalize "indole-3-butyric acid" here and in other places (also this concerns indole-3-acetic acid).

Abstract

Do not use abbreviation for plant growth regulators in the Abstract, as the term is used only once. First define term in full and then provide abbreviation in parentheses. Do not use ppm as unit of measurement, use real units instead (as mg per L or as appropriate), here and throughout the manuscript.

Do not use abbreviated terms as keywords (PGRs). Instead of name of specific computer software (WinRHIZO) use "image analysis".

Introduction

In general, introduction is not appropriate. It represents collection of somehow unrelated paragraphs or even separate sentences without any logical structure. It needs to be completely rewritten, starting with broader practical aspects of woody plant vegetative propagation, followed by particular methods and problems related to their implementation as well as analysis of functional aspects. It is somehow misleading to use term "auxins, small organic molecules" in respect to "phytohormones", as there is only one generally accepted natural auxin (IAA).

I did not get context why detailed paragraph on Xylella fastidiosa is included. Is it used to justify the need for propagation methods of resistant ornamental woody plants? Then it is necessary to clearly indicate so.

Do not italicise "taxa" (line 108, also line 308).

What is "K-IBA" (line 113)?

Results

It is advised to include all results from cultivar vs IBA concentration combinations with their statistical evaluation in Tables 1, 2, 4, 5, 6 and 7 instead of mean values of both cultivars at each concentration and mean values for cultivar across all concentration range, similar to as it is already done in Table 3. Figures 1 and 2 lack scale bar, it seems that different scales are used in individual photos. In general, it is not a good practice to use mounted images for comparison of treatments. Photo "IBA 2.500 ppm cv2" in Figure 2 shows signs of selective background removal, which is not acceptable practice in scientific publications.

There is a lack of indication of statistical significance of differences in several parts of Figures 5 and 6.

Discussion is overly concise and descriptive. There is no general problem-oriented discussion, only some facts from the present study are compared with literature data in a point-by-point manner (using author-oriented style of citation, which seems to be inappropriate). A clear message in respect to both practical and functional outcomes from the present study needs to appear here. It was not necessary to perform the study, if the general conclusion is only as follows: "Our study found that cultivars from both taxa respond differently to different IBA concentrations". I agree that there is dedicated conclusions part, but then rather trivial last paragraph from discussion is redundant. In respect to Conclusions, I need to point out that the last paragraph better suits discussion, as no market studies were performed within the present study.

Materials and methods

Do not capitalize "carbon" and "nitrogen". Do not use "=" sign (lines 353–354).

Use space between numbers and units of measurement (00 °C).

List of references contains many errors and style inconsequences.

Author Response

The authors rewrote large portions of the manuscript

  • Introduction was improved and it provides sufficient background. It now includes more relevant reference
  • Results and discussion were improved. They are clearly presented.
  • Conclusions were improved. They are supported by the results

Title

Do not capitalize "indole-3-butyric acid" here and in other places (also this concerns indole-3-acetic acid).

Done

Abstract

Do not use abbreviation for plant growth regulators in the Abstract, as the term is used only once. First define term in full and then provide abbreviation in parentheses. Do not use ppm as unit of measurement, use real units instead (as mg per L or as appropriate), here and throughout the manuscript.

Done, abstract was improved

Do not use abbreviated terms as keywords (PGRs). Done: line 32

Instead of name of specific computer software (WinRHIZO) use "image analysis". Image analisys has been deleted from the keywords because it is already reported in the title. We have added woody plants: line 33

Introduction

In general, introduction is not appropriate. It represents collection of somehow unrelated paragraphs or even separate sentences without any logical structure. It needs to be completely rewritten, starting with broader practical aspects of woody plant vegetative propagation, followed by particular methods and problems related to their implementation as well as analysis of functional aspects.

Line 36-53

 It is somehow misleading to use term "auxins, small organic molecules" in respect to "phytohormones", as there is only one generally accepted natural auxin (IAA).

The authors discussed phytohormones

I did not get context why detailed paragraph on Xylella fastidiosa is included. Is it used to justify the need for propagation methods of resistant ornamental woody plants? Then it is necessary to clearly indicate so.

This paragraph has been summarized. Lantana and Abelia taxa are not host plants of the bacterium, so this paragraph is used to justify the need for propagation methods. Lines 69-71

Do not italicise "taxa" (line 108, also line 308). Done

What is "K-IBA" (line 113)? Potassium salt of IBA, line 90

Results

It is advised to include all results from cultivar vs IBA concentration combinations with their statistical evaluation in Tables 1, 2, 4, 5, 6 and 7 instead of mean values of both cultivars at each concentration and mean values for cultivar across all concentration range, similar to as it is already done in Table 3.

Done: Table 1 (line109), Table 2 (line 161), Table 3 (line 179), Table 4 (line 201) and Table 5 8line 217).

Figures 1 and 2 lack scale bar, it seems that different scales are used in individual photos. In general, it is not a good practice to use mounted images for comparison of treatments. Photo "IBA 2.500 ppm cv2" in Figure 2 shows signs of selective background removal, which is not acceptable practice in scientific publications.

Figures 1 and 2 were deleted

There is a lack of indication of statistical significance of differences in several parts of Figures 5 and 6.

Ex Figures 5 and 6, now Figure 3 (line 227) , Figure 3 was improved with indication of statistical significance of differences

Discussion is overly concise and descriptive. There is no general problem-oriented discussion, only some facts from the present study are compared with literature data in a point-by-point manner (using author-oriented style of citation, which seems to be inappropriate). A clear message in respect to both practical and functional outcomes from the present study needs to appear here.

The authors have highlighted the importance of auxin use in improving AR formation and maximizing production efficiency in woody species with high market demand

It was not necessary to perform the study, if the general conclusion is only as follows: "Our study found that cultivars from both taxa respond differently to different IBA concentrations". I agree that there is dedicated conclusions part, but then rather trivial last paragraph from discussion is redundant.

In respect to Conclusions, I need to point out that the last paragraph better suits discussion, as no market studies were performed within the present study. Last paragraph was deleted

The paragraph ‘Conclusions’ was been rewritten following the suggestions 

Materials and methods

Do not capitalize "carbon" and "nitrogen". Done: lines 309-310

Do not use "=" sign (lines 353–354). Done

Use space between numbers and units of measurement (00 °C). Done

List of references contains many errors and style inconsequences. List of references was improved.

Round 2

Reviewer 2 Report

The manuscript has been improved significantly. Nevertheless, there remain some open question and remarks.

l. 40 sexual reproduction instead of "reproduction"

l. 120 "root length" means total root length? This should be added. The same is true for Fig. 1

M+M: How many cuttings did you take per mother tree? And are the mother trees ramets or different genotypes? This information should be given to the readers.

l. 208: was shown instead of "was showed"

l.248: including instead of "included"

l. 272: roots per plant? instead of only "roots"

l. 275: genotype instead of "genetic component"

l.337: (roots>0,5 cm length)?

The authors give no information about standard deviation, but it's important for the reader to get an impression on the deviation of the data.

Author Response

Bari, January 17, 2022

Dear reviewer,

Manuscript was improved accepting the your remarks and providing a response to all comments.

Methods: authors added this sentence: For this study, 18 median cuttings were taken randomly from a single clonal mother plant, in each taxon and cultivar.

Results: In Tables and Figures, the authors provided information about the standard error.
